# Variational Walkback: Learning a Transition Operator as a Stochastic Recurrent Net

**Anirudh Goyal**
MILA, Université de Montréal
anirudhgoyal9119@gmail.com

**Nan Rosemary Ke**
MILA, École Polytechnique de Montréal
rosemary.nan.ke@gmail.com

**Surya Ganguli**
Stanford University
sganguli@stanford.edu

**Yoshua Bengio**
MILA, Université de Montréal
yoshua.umontreal@gmail.com

## Abstract

We propose a novel method to *directly* learn a stochastic transition operator whose repeated application provides generated samples. Traditional undirected graphical models approach this problem indirectly by learning a Markov chain model whose stationary distribution obeys detailed balance with respect to a parameterized energy function. The energy function is then modified so the model and data distributions match, with no guarantee on the number of steps required for the Markov chain to converge. Moreover, the detailed balance condition is highly restrictive: energy based models corresponding to neural networks must have symmetric weights, unlike biological neural circuits. In contrast, we develop a method for directly learning arbitrarily parameterized transition operators capable of expressing non-equilibrium stationary distributions that violate detailed balance, thereby enabling us to learn more biologically plausible asymmetric neural networks and more general non-energy based dynamical systems. The proposed training objective, which we derive via principled variational methods, encourages the transition operator to "walk back" (prefer to revert its steps) in multi-step trajectories that start at data-points, as quickly as possible back to the original data points. We present a series of experimental results illustrating the soundness of the proposed approach, Variational Walkback (VW), on the MNIST, CIFAR-10, SVHN and CelebA datasets, demonstrating superior samples compared to earlier attempts to learn a transition operator. We also show that although each rapid training trajectory is limited to a finite but variable number of steps, our transition operator continues to generate good samples well past the length of such trajectories, thereby demonstrating the match of its non-equilibrium stationary distribution to the data distribution. Source Code: http://github.com/anirudh9119/walkback_nips17

## 1 Introduction

A fundamental goal of unsupervised learning involves training generative models that can understand sensory data and employ this understanding to generate, or sample new data and make new inferences. In machine learning, the vast majority of probabilistic generative models that can learn complex probability distributions over data fall into one of two classes: (1) directed graphical models, corresponding to a finite time feedforward generative process (e.g. variants of the Helmholtz machine (Dayan et al., 1995) like the Variational Auto-Encoder (VAE) (Kingma and Welling, 2013; Rezende et al., 2014)), or (2) energy function based undirected graphical models, corresponding to sampling from a stochastic process whose *equilibrium* stationary distribution obeys detailed balance with respect to the energy function (e.g. various Boltzmann machines (Salakhutdinov and Hinton, 2009)). This detailed

balance condition is highly restrictive: for example, energy-based undirected models corresponding to neural networks require symmetric weight matrices and very specific computations which may not match well with what biological neurons or analog hardware could compute.

In contrast, biological neural circuits are capable of powerful generative dynamics enabling us to model the world and imagine new futures. Cortical computation is highly recurrent and therefore its generative dynamics cannot simply map to the purely feed-forward, finite time generative process of a directed model. Moreover, the recurrent connectivity of biological circuits is not symmetric, and so their generative dynamics cannot correspond to sampling from an energy-based undirected model.

Thus, the asymmetric biological neural circuits of our brain instantiate a type of stochastic dynamics arising from the repeated application of a transition operator* whose stationary distribution over neural activity patterns is a *non-equilibrium* distribution that does not obey detailed balance with respect to any energy function. Despite these fundamental properties of brain dynamics, machine learning approaches to training generative models currently lack effective methods to model complex data distributions through the repeated application a transition operator, that is not indirectly specified through an energy function, but rather is *directly* parameterized in ways that are inconsistent with the existence of *any* energy function. Indeed the lack of such methods constitutes a glaring gap in the pantheon of machine learning methods for training probabilistic generative models.

The fundamental goal of this paper is to provide a step to filling such a gap by proposing a novel method to learn such directly parameterized transition operators, thereby providing an empirical method to control the stationary distributions of non-equilibrium stochastic processes that do not obey detailed balance, and match these distributions to data. The basic idea underlying our training approach is to start from a training example, and iteratively apply the transition operator while gradually increasing the amount of noise being injected (i.e., temperature). This heating process yields a trajectory that starts from the data manifold and walks away from the data due to the heating and to the mismatch between the model and the data distribution. Similarly to the update of a denoising autoencoder, we then modify the parameters of the transition operator so as to make the *reverse* of this heated trajectory *more* likely under a reverse cooling schedule. This encourages the transition operator to generate stochastic trajectories that evolve towards the data distribution, by learning to walk back the heated trajectories starting at data points. This walkback idea had been introduced for generative stochastic networks (GSNs) and denoising autoencoders (Bengio et al., 2013b) as a heuristic, and without temperature annealing. Here, we derive the specific objective function for learning the parameters through a principled variational lower bound, hence we call our training method variational walkback (VW). Despite the fact that the training procedure involves walking back a set of trajectories that last a finite, but variable number of time-steps, we find empirically that this yields a transition operator that continues to generate sensible samples for many more time-steps than are used to train, demonstrating that our finite time training procedure can sculpt the non-equilibrium stationary distribution of the transition operator to match the data distribution.

We show how VW emerges naturally from a variational derivation, with the need for annealing arising out of the objective of making the variational bound as tight as possible. We then describe experimental results illustrating the soundness of the proposed approach on the MNIST, CIFAR-10, SVHN and CelebA datasets. Intriguingly, we find that our finite time VW training process involves modifications of variational methods for training directed graphical models, while our potentially asymptotically infinite generative sampling process corresponds to non-equilibrium generalizations of energy based undirected models. Thus VW goes beyond the two disparate model classes of undirected and directed graphical models, while simultaneously incorporating good ideas from each.

## 2 The Variational Walkback Training Process

Our goal is to learn a stochastic transition operator $p_T(s'|s)$ such that its repeated application yields samples from the data manifold. Here $T$ reflects an underlying temperature, which we will modify during the training process. The transition operator is further specified by other parameters which must be learned from data. When $K$ steps are chosen to generate a sample, the generative process has joint probability $p(s_0^K) = p(s_K) \prod_{t=1}^{K} p_{T_t}(s_{t-1}|s_t)$, where $T_t$ is the temperature at step $t$. We first give an intuitive description of our learning algorithm before deriving it via variational methods in the next section. The basic idea, as illustrated in Fig. 1 and Algorithm 1 is to follow a walkback

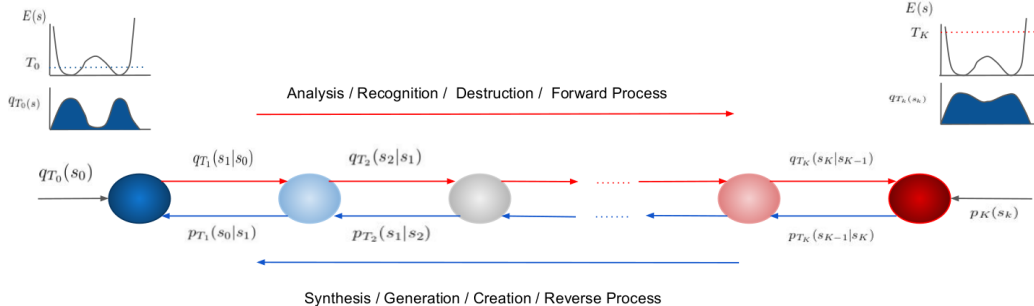

Figure 1: Variational WalkBack framework. The generative process is represented in the blue arrows with the sequence of $p_{T_t}(\boldsymbol{s}_{t-1}|\boldsymbol{s}_t)$ transitions. The destructive forward process starts at a datapoint (from $q_{T_0}(\boldsymbol{s}_0)$) and gradually heats it through applications of $q_{T_t}(\boldsymbol{s}_t|\boldsymbol{s}_{t-1})$. Larger temperatures on the right correspond to a flatter distribution, so the whole destructive forward process maps the data distribution to a Gaussian and the creation process operates in reverse.

strategy similar to that introduced in Alain and Bengio (2014). In particular, imagine a destructive process $q_{T_{t+1}}(\boldsymbol{s}_{t+1}|\boldsymbol{s}_t)$ (red arrows in Fig. 1), which starts from a data point $\boldsymbol{s}_0 = \boldsymbol{x}$, and evolves it stochastically to obtain a trajectory $\boldsymbol{s}_0, \ldots, \boldsymbol{s}_K \equiv \boldsymbol{s}_0^K$, i.e., $q(\boldsymbol{s}_0^K) = q(s_0) \prod_{t=1}^{K} q_{T_t}(s_t|s_{t-1})$, where $q(s_0)$ is the data distribution. Note that the $p$ and $q$ chains will share the same parameters for the transition operator (one going backwards and one forward) but they start from different priors for their first step: $q(s_0)$ is the data distribution while $p(s_0)$ is a flat factorized prior (e.g. Gaussian). The training procedure trains the transition operator $p_T$ to make reverse transitions of the destructive process more likely. For this reason we index time so the destructive process operates forward in time, while the reverse generative process operates backwards in time, with the data distribution occurring at $t = 0$. In particular, we need only train the transition operator to reverse time by 1-step at each step, making it unnecessary to solve a deep credit assignment problem by performing backpropagation through time across multiple walk-back steps. Overall, the destructive process generates trajectories that walk away from the data manifold, and the transition operator $p_T$ learns to walkback these trajectories to sculpt the stationary distribution of $p_T$ at $T = 1$ to match the data distribution.

Because we choose $q_T$ to have the *same* parameters as $p_T$, they have the same transition operator but not the same joint over the whole sequence because of differing initial distributions for each trajectory. We also choose to increase temperature with time in the destructive process, following a temperature schedule $T_1 \leq \cdots \leq T_K$. Thus the forward destructive (reverse generative) process corresponds to a heating (cooling) protocol. This training procedure is similar in spirit to DAE's (Vincent et al., 2008) or NET (Sohl-Dickstein et al., 2015) but with one major difference: the destructive process in these works corresponds to the addition of random noise which knows nothing about the current generative process during training. To understand why tying together destruction and creation may be a good idea, consider the special case in which $p_T$ corresponds to a stochastic process whose stationary distribution obeys detailed balance with respect to the energy function of an undirected graphical model. Learning any such model involves two fundamental goals: the model must place probability mass (i.e. lower the energy function) where the data is located, and remove probability mass (i.e. raise the energy function) elsewhere. Probability modes where there is no data are known as spurious modes, and a fundamental goal of learning is to hunt down these spurious modes and remove them. Making the destructive process *identical* to the transition operator to be learned is motivated by the notion that the destructive process should then efficiently explore the spurious modes of the current transition operator. The walkback training will then destroy these modes. In contrast, in DAE's and NET's, since the destructive process corresponds to the addition of unstructured noise that knows nothing about the generative process, it is not clear that such an agnostic destructive process will efficiently seek out the spurious modes of the reverse, generative process.

We chose the annealing schedule empirically to minimize training time. The generative process starts by sampling a state $\boldsymbol{s}_K$ from a broad Gaussian $p^*(\boldsymbol{s}_K)$, whose variance is initially equal to the total data variance $\sigma_{\max}^2$ (but can be later adapted to match the final samples from the inference trajectories). Then we sample from $p_{T_{\max}}(\boldsymbol{s}_{K-1}|\boldsymbol{s}_K)$, where $T_{\max}$ is a high enough temperature so that the resultant injected noise can move the state across the whole domain of the data. The injected noise used to simulate the effects of finite temperature has variance linearly proportional to

temperature. Thus if $\sigma^2$ is the equivalent noise injected by the transition operator $p_T$ at $T = 1$, we choose $T_{\max} = \frac{\sigma^2_{\max}}{\sigma^2}$ to achieve the goal of the first sample $s_{K-1}$ being able to move across the entire range of the data distribution. Then we successively cool the temperature as we sample "previous" states $s_{t-1}$ according to $p_T(s_{t-1}|s_t)$, with $T$ reduced by a factor of 2 at each step, followed by $n$ steps at temperature 1. This cooling protocol requires the number of steps to be

$$K = \log_2 T_{\max} + n, \tag{1}$$

in order to go from $T = T_{\max}$ to $T = 1$ in $K$ steps. We choose $K$ from a random distribution. Thus the training procedure trains $p_T$ to rapidly transition from a simple Gaussian distribution to the data distribution in a finite but variable number of steps. Ideally, this training procedure should then indirectly create a transition operator $p_T$ at $T = 1$ whose repeated iteration samples the data distribution with a relatively rapid mixing time. Interestingly, this intuitive learning algorithm for a recurrent dynamical system, formalized in Algorithm 1, can be derived in a principled manner from variational methods that are usually applied to directed graphical models, as we see next.

---

**Algorithm 1 VariationalWalkback($\boldsymbol{\theta}$)**

Train a generative model associated with a transition operator $p_T(s|s')$ at temperature $T$ (temperature 1 for sampling from the actual model), parameterized by $\boldsymbol{\theta}$. This transition operator injects noise of variance $T\sigma^2$ at each step, where $\sigma^2$ is the noise level at temperature 1.

---

**Require:** Transition operator $p_T(s|s')$ from which one can both sample and compute the gradient of $\log p_T(s|s')$ with respect to parameters $\theta$, given $s$ and $s'$.
**Require:** Precomputed $\sigma^2_{\max}$, initially data variance (or squared diameter).
**Require:** $N_1 > 1$ the number of initial temperature-1 steps of $q$ trajectory (or ending a $p$ trajectory).
  **repeat**
    Set $p^*$ to be a Gaussian with mean and variance of the data.
    $T_{\max} \leftarrow \frac{\sigma^2_{\max}}{\sigma^2}$
    Sample $n$ as a uniform integer between 0 and $N_1$
    $K \leftarrow \text{ceil}(\log_2 T_{\max}) + n$
    Sample $x \sim$ data (or equivalently sample a minibatch to parallelize computation and process each element of the minibatch independently)
    Let $s_0 = (x)$ and initial temperature $T = 1$, initialize $\mathcal{L} = 0$
    **for** $t = 1$ to $K$ **do**
      Sample $s_t \sim p_T(s|s_{t-1})$
      Increment $\mathcal{L} \leftarrow \mathcal{L} + \log p_T(s_{t-1}|s_t)$
      Update parameters with log likelihood gradient $\frac{\partial \log p_T(s_{t-1}|s_t)}{\partial \theta}$
      If $t > n$, increase temperature with $T \leftarrow 2T$
    **end for**
    Increment $\mathcal{L} \leftarrow \mathcal{L} + \log p^*(s_K)$
    Update mean and variance of $p^*$ to match the accumulated 1st and 2nd moment statistics of the samples of $s_K$
  **until** convergence monitoring $\mathcal{L}$ on a validation set and doing early stopping =0

---

## 3 Variational Derivation of Walkback

The marginal probability of a data point $s_0$ at the end of the $K$-step generative cooling process is

$$p(s_0) = \sum_{s_1^K} p_{T_0}(s_0|s_1) \left( \prod_{t=2}^{K} p_{T_t}(s_{t-1}|s_t) \right) p^*(s_K) \tag{2}$$

where $s_1^K = (s_1, s_2, \ldots, s_K)$ and $v = s_0$ is a visible variable in our generative process, while the cooling trajectory that lead to it can be thought of as a latent, hidden variable $h = s_1^K$. Recall the decomposition of the marginal log-likelihood via a variational lower bound,

$$\ln p(v) \equiv \ln \sum_h p(v|h)p(h) = \underbrace{\sum_h q(h|v) \ln \frac{p(v,h)}{q(h|v)}}_{\mathcal{L}} + D_{KL}[q(h|v)||p(h|v)]. \tag{3}$$

Here $\mathcal{L}$ is the variational lower bound which motivates the proposed training procedure, and $q(h|v)$ is a variational approximation to $p(h|v)$. Applying this decomposition to $\boldsymbol{v} = \boldsymbol{s}_0$ and $\boldsymbol{h} = \boldsymbol{s}_1^K$, we find

$$\ln p(s_0) = \sum_{s_1^k} q(s_1^k|s_0) \ln \frac{p(s_0|s_1^k)p(s_1^k)}{q(s_1^k|s_0)} + D_{KL}[q(s_1^k|s_0) \,||\, p(s_1^k|s_0)]. \tag{4}$$

Similarly to the EM algorithm, we aim to approximately maximize the log-likelihood with a 2-step procedure. Let $\theta_p$ be the parameters of the generative model $p$ and $\theta_q$ be the parameters of the approximate inference procedure $q$. Before seeing the next example we have $\theta_q = \theta_p$. Then in the first step we update $\theta_p$ towards maximizing the variational bound $\mathcal{L}$, for example by a stochastic gradient descent step. In the second step, we update $\theta_q$ by setting $\theta_q \leftarrow \theta_p$, with the objective to reduce the KL term in the above decomposition. See Sec. 3.1 below regarding conditions for the tightness of the bound, which may not be perfect, yielding a possibly biased gradient when we force the constraint $\theta_p = \theta_q$. We continue iterating this procedure, with training examples $s^0$. We can obtain an unbiased Monte-Carlo estimator of $\mathcal{L}$ as follows from a single trajectory:

$$\mathcal{L}(s^0) \approx \sum_{t=1}^{K} \ln \frac{p_{T_t}(s_{t-1}|s_t)}{q_{T_t}(s_t|s_{t-1})} + \ln p^*(s_K) \tag{5}$$

with respect to $p_\theta$, where $s^0$ is sampled from the data distribution $q_{T_0}(s^0)$, and the single sequence $s_1^K$ is sampled from the heating process $q(s_1^K|s_0)$. We are making the reverse of heated trajectories more likely under the cooling process, leading to Algorithm 1. Such variational bounds have been used successfully in many learning algorithms in the past, such as the VAE (Kingma and Welling, 2013), except that they use an explicitly different set of parameters for $p$ and $q$. Some VAE variants (Sønderby et al., 2016; Kingma et al., 2016) however mix the $p$-parameters implicitly in forming $q$, by using the likelihood gradient to iteratively form the approximate posterior.

## 3.1 Tightness of the variational lower bound

As seen in (4), the gap between $\mathcal{L}(s_0)$ and $\ln p(s_0)$ is controlled by $D_{KL}[q(s_1^k|s_0)||p(s_1^k|s_0)]$, and is therefore tight when the distribution of the heated trajectory, *starting* from a point $s_0$, matches the posterior distribution of the cooled trajectory *ending* at $s_0$. Explicitly, this KL divergence is given by

$$D_{KL} = \sum_{s_1^k} q(s_1^k|s_0) \ln \frac{p(s_0)}{p^*(s_K)} \prod_{t=1}^{K} \frac{q_{T_t}(s_t|s_{t-1})}{p_{T_t}(s_{t-1}|s_t)}. \tag{6}$$

As the heating process $q$ unfolds forward in time, while the cooling process $p$ unfolds backwards in time, we introduce the *time reversal* of the transition operator $p_T$, denoted by $p_T^R$, as follows. Under repeated application of the transition operator $p_T$, state $s$ settles into a stationary distribution $\pi_T(s)$ at temperature $T$. The probability of observing a transition $s_t \rightarrow s_{t-1}$ under $p_T$ in its stationary state is then $p_T(s_{t-1}|s_t)\pi_T(s_t)$. The time-reversal $p_T^R$ is the transition operator that makes the reverse transition equally likely for all state pairs, and therefore obeys

$$P_T(s_{t-1}|s_t)\pi_T(s_t) = P_T^R(s_t|s_{t-1})\pi_T(s_{t-1}) \tag{7}$$

for all pairs of states $s_{t-1}$ and $s_t$. It is well known that $p_T^R$ is a valid stochastic transition operator and has the same stationary distribution $\pi_T(s)$ as $p_T$. Furthermore, the process $p_T$ obeys detailed balance if and only if it is invariant under time-reversal, so that $p_T = p_T^R$.

To better understand the KL divergence in (6), at each temperature $T_t$, we use relation (7) to replace the cooling process $P_{T_t}$ which occurs backwards in time with its time-reversal, unfolding forward in time, at the expense of introducing ratios of stationary probabilities. We also exploit the fact that $q$ and $p$ are the same transition operator. With these substitutions in (6), we find

$$D_{KL} = \sum_{s_1^k} q(s_1^k|s_0) \ln \prod_{t=1}^{K} \frac{p_{T_t}(s_t|s_{t-1})}{p_{T_t}^R(s_t|s_{t-1})} + \sum_{s_1^k} q(s_1^k|s_0) \ln \frac{p(s_0)}{p^*(s_K)} \prod_{t=1}^{K} \frac{\pi_{T_t}(s_t)}{\pi_{T_t}(s_{t-1})}. \tag{8}$$

The first term in (8) is simply the KL divergence between the distribution over heated trajectories, and the time reversal of the cooled trajectories. Since the heating ($q$) and cooling ($p$) processes are tied, this KL divergence is 0 if and only if $p_{T_t} = p_{T_t}^R$ for all $t$. This time-reversal invariance requirement for vanishing KL divergence is equivalent to the transition operator $p_T$ obeying detailed balance at all temperatures.

Now intuitively, the second term can be made small in the limit where $K$ is large and the temperature sequence is annealed slowly. To see why, note we can write the ratio of probabilities in this term as,

$$\frac{p(s_0)}{\pi_{T_1}(s_0)} \frac{\pi_{T_1}(s_1)}{\pi_{T_2}(s_1)} \cdots \frac{\pi_{T_{K-1}}(s_{K-1})}{\pi_{T_{K-1}}(s_K)} \frac{\pi_{T_K}(s_K)}{p^*(s_K)}. \tag{9}$$

which is similar in shape (but arising in a different context) to the product of probability ratios computed for annealed importance sampling (Neal, 2001) and reverse annealed importance sampling (Burda et al., 2014). Here it is manifest that, under slow incremental annealing schedules, we are comparing probabilities of the same state under slightly different distributions, so all ratios are close to 1. For example, under many steps, with slow annealing, the generative process approximately reaches its stationary distribution, $p(s_0) \approx \pi_{T_1}(s_0)$.

This slow annealing to go from $p^*(s_K)$ to $p(s_0)$ corresponds to the quasistatic limit in statistical physics, where the work required to perform the transformation is equal to the free energy difference between states. To go faster, one must perform excess work, above and beyond the free energy difference, and this excess work is dissipated as heat into the surrounding environment. By writing the distributions in terms of energies and free energies: $\pi_{T_t}(s_t) \propto e^{-E(s_t)/T_t}$, $p^*(s_K) = e^{-[E_K(s_K)-F_K]}$, and $p(s_0) = e^{-[E_0(s_0)-F_0]}$, one can see that the second term in the KL divergence is closely related to average heat dissipation in a finite time heating process (see e.g. (Crooks, 2000)).

This intriguing connection between the size of the gap in a variational lower bound, and the excess heat dissipation in a finite time heating process opens the door to exploiting a wealth of work in statistical physics for finding optimal thermodynamic paths that minimize heat dissipation (Schmiedl and Seifert, 2007; Sivak and Crooks, 2012; Gingrich et al., 2016), which may provide new ideas to improve variational inference. In summary, tightness of the variational bound can be achieved if: (1) The transition operator of $p$ approximately obeys detailed balance, and (2) the temperature annealing is done slowly over many steps. And intriguingly, the magnitude of the looseness of the bound is related to two physical quantities: (1) the degree of irreversiblity of the transition operator $p$, as measured by the KL divergence between $p$ and its *time reversal* $p^R$, and (2) the excess physical work, or equivalently, excess heat dissipated, in performing the heating trajectory.

To check, post-hoc, potential looseness of the variational lower bound, we can measure the degree of irreversibility of $p_T$ by estimating the KL divergence $D_{KL}(p_T(s'|s)\pi_T(s) \,||\, p_T(s|s')\pi_T(s'))$, which is 0 if and only if $p_T$ obeys detailed balance and is therefore time-reversal invariant. This quantity can be estimated by $\frac{1}{K}\sum_{t=1}^{K} \ln \frac{p_T(s_{t+1}|s_t)}{p_T(s_t|s_{t+1})}$, where $s_1^K$ is a long sequence sampled by repeatedly applying transition operator $p_T$ from a draw $s_1 \sim \pi_T$. If this quantity is strongly positive (negative) then forward transitions are more (less) likely than reverse transitions, and the process $p_T$ is not time-reversal invariant. This estimated KL divergence can be normalized by the corresponding entropy to get a relative value (with 3.6% measured on a trained model, as detailed in Appendix).

### 3.2 Estimating log likelihood via importance sampling

We can derive an importance sampling estimate of the negative log-likelihood by the following procedure. For each training example $x$, we sample a large number of destructive paths (as in Algorithm 1). We then use the following formulation to estimate the log-likelihood $\log p(x)$ via

$$\log \mathbb{E}_{x \sim p_{\mathcal{D}}, q_{T_0}(x)q_{T_1}(s_1|s_0(x,))\left(\prod_{t=2}^{K} q_{T_t}(s_t|s_{t-1})\right)} \left[ \frac{p_{T_0}(s_0 = x|s_1)\left(\prod_{t=2}^{K} p_{T_t}(s_{t-1}|s_t)\right) p^*(s_K)}{q_{T_0}(x)q_{T_1}(s_1|s_0 = x)\left(\prod_{t=2}^{K} q_{T_t}(s_t|s_{t-1})\right)} \right] \tag{10}$$

### 3.3 VW transition operators and their convergence

The VW approach allows considerable freedom in choosing transition operators, obviating the need for specifying them indirectly through an energy function. Here we consider Bernoulli and isotropic Gaussian transition operators for binary and real-valued data respectively. The form of the stochastic state update imitates a discretized version of the Langevin differential equation. The Bernoulli transition operator computes the element-wise probability as $\rho = \text{sigmoid}(\frac{(1-\alpha)*s_{t-1}+\alpha*F_\rho(s_{t-1})}{T_t})$. The Gaussian operator computes a conditional mean and standard deviation via $\mu = (1-\alpha)*s_{t-1} + \alpha*F_\mu(s_{t-1})$ and $\sigma = T_t \log(1+e^{F_\sigma(s_{t-1})})$. Here the $F$ functions can be arbitrary parametrized functions, such as a neural net and $T_t$ is the temperature at time step t.

A natural question is when will the finite time VW training process learn a transition operator whose stationary distribution matches the data distribution, so that repeated sampling far beyond the training time continues to yield data samples. To partially address this, we prove the following theorem:

**Proposition 1.** *If $p$ has enough capacity, training data and training time, with slow enough annealing and a small departure from reversibility so $p$ can match q, then at convergence of VW training, the transition operator $p_T$ at $T = 1$ has the data generating distribution as its stationary distribution.*

A proof can be found in the Appendix, but the essential intuition is that if the finite time generative process converges to the data distribution at multiple different VW walkback time-steps, then it remains on the data distribution for all future time at $T = 1$. We cannot always guarantee the preconditions of this theorem but we find experimentally that its essential outcome holds in practice.

## 4 Related Work

A variety of learning algorithms can be cast in the framework of Fig. 1. For example, for directed graphical models like VAEs (Kingma and Welling, 2013; Rezende et al., 2014), DBNs (Hinton et al., 2006), and Helmholtz machines in general, $q$ corresponds to a recognition model, transforming data to a latent space, while $p$ corresponds to a generative model that goes from latent to visible data in a finite number of steps. None of these directed models are designed to learn transition operators that can be iterated *ad infinitum*, as we do. Moreover, learning such models involves a complex, deep credit assignment problem, limiting the number of unobserved latent layers that can be used to generate data. Similar issues of limited trainable depth in a finite time feedforward generative process apply to Generative Adversarial Networks (GANs) (Goodfellow et al., 2014), which also further eschew the goal of specifically assigning probabilities to data points. Our method circumvents this deep credit assignment problem by providing training targets at each time-step; in essence each past time-step of the heated trajectory constitutes a training target for the future output of the generative operator $p_T$, thereby obviating the need for backpropagation across multiple steps. Similarly, unlike VW, Generative Stochastic Networks (GSN) (Bengio et al., 2014) and the DRAW (Gregor et al., 2015) also require training iterative operators by backpropagating across multiple computational steps.

VW is similar in spirit to DAE (Bengio et al., 2013b), and NET approaches (Sohl-Dickstein et al., 2015) but it retains two crucial differences. First, in each of these frameworks, $q$ corresponds to a very simple destruction process in which unstructured Gaussian noise is injected into the data. This agnostic destruction process has no knowledge of underlying generative process $p$ that is to be learned, and therefore cannot be expected to efficiently explore spurious modes, or regions of space, unoccupied by data, to which $p$ assigns high probability. VW has the advantage of using a high-temperature version of the model $p$ itself as part of the destructive process, and so should be better than random noise injection at finding these spurious modes. A second crucial difference is that VW ties weights of the transition operator across time-steps, thereby enabling us to learn a *bona fide* transition operator than can be iterated well beyond the training time, unlike DAEs and NET. There's also another related recent approach to learning a transition operator with a denoising cost, developed in parallel, called Infusion training (Bordes et al., 2017), which tries to reconstruct the target data in the chain, instead of the previous step in the destructive chain.

## 5 Experiments

VW is evaluated on four datasets: MNIST, CIFAR10 (Krizhevsky and Hinton, 2009), SVHN (Netzer et al., 2011) and CelebA (Liu et al., 2015). The MNIST, SVHN and CIFAR10 datasets were used as is except for uniform noise added to MNIST and CIFAR10, as per Theis et al. (2016), and the aligned and cropped version of CelebA was scaled from 218 x 178 pixels to 78 x 64 pixels and center-cropped at 64 x 64 pixels (Liu et al., 2015). We used the Adam optimizer (Kingma and Ba, 2014) and the Theano framework (Al-Rfou et al., 2016). More details are in Appendix and code for training and generation is at `http://github.com/anirudh9119/walkback_nips17`.

Table 1 compares with published NET results on CIFAR.

**Image Generation.** Figure 3, 5, 6, 7, 8 (see supplementary section) show VW samples on each of the datasets. For MNIST, real-valued views of the data are modeled. **Image Inpainting.** We clamped the bottom part of CelebA test images (for each step during sampling), and ran it through the model. Figure 1 (see Supplementary section) shows the generated conditional samples.

| Model | bits/dim $\leq$ |
|---|---|
| NET  (Sohl-Dickstein et al., 2015) | 5.40 |
| VW(20 steps) | 5.20 |
| Deep VAE | < 4.54 |
| VW(30 steps) | 4.40 |
| DRAW (Gregor et al., 2015) | < 4.13 |
| ResNet VAE with IAF  (Kingma et al., 2016) | 3.11 |

Table 1: Comparisons on CIFAR10, test set average number of bits/data dimension(lower is better)

# 6   Discussion

## 6.1   Summary of results

Our main advance involves using variational inference to learn recurrent transition operators that can rapidly approach the data distribution and then be iterated much longer than the training time while still remaining on the data manifold. Our innovations enabling us to achieve this involved: (a) tying weights across time, (b) tying the destruction and generation process together to efficiently destroy spurious modes, (c) using the past of the destructive process to train the future of the creation process, thereby circumventing issues with deep credit assignment (like NET), (d) introducing an aggressive temperature annealing schedule to rapidly approach the data distribution (e.g. NET takes 1000 steps while VWB only takes 30 steps to do so), and (e) introducing variable trajectory lengths during training to encourage the generator to stay on the data manifold for times longer than the training sequence length.

Indeed, it is often difficult to sample from recurrent neural networks for many more time steps than the duration of their training sequences, especially non-symmetric networks that could exhibit chaotic activity. Transition operators learned by VW can be stably sampled for exceedingly long times; for example, in experiments (see supplementary section) we trained our model on CelebA for 30 steps, while at test time we sampled for 100000 time-steps. Overall, our method of learning a transition operator outperforms previous attempts at learning transition operators (i.e. VAE, GSN and NET) using a local learning rule.

Overall, we introduced a new approach to learning non-energy-based transition operators which inherits advantages from several previous generative models, including a training objective that requires rapidly generating the data in a finite number of steps (as in directed models), re-using the same parameters for each step (as in undirected models), directly parametrizing the generator (as in GANs and DAEs), and using the model itself to quickly find its own spurious modes (the walk-back idea). We also anchor the algorithm in a variational bound and show how its analysis suggests to use the same transition operator for the destruction or inference process, and the creation or generation process, and to use a cooling schedule during generation, and a reverse heating schedule during inference.

## 6.2   New bridges between variational inference and non-equilibrium statistical physics

We connected the variational gap to physical notions like reversibility and heat dissipation. This novel bridge between variational inference and concepts like excess heat dissipation in non-equilbrium statistical physics, could potentially open the door to improving variational inference by exploiting a wealth of work in statistical physics. For example, physical methods for finding optimal thermodynamic paths that minimize heat dissipation (Schmiedl and Seifert, 2007; Sivak and Crooks, 2012; Gingrich et al., 2016), could potentially be exploited to tighten lowerbounds in variational inference. Moreover, motivated by the relation between the variational gap and reversibility, we verified empirically that the model converges towards an *approximately* reversible chain (see Appendix) making the variational bound tighter.

## 6.3   Neural weight asymmetry

A fundamental aspect of our approach is that we can train stochastic processes that need not exactly

obey detailed balance, yielding access to a larger and potentially more powerful space of models. In particular, this enables us to relax the weight symmetry constraint of undirected graphical models corresponding to neural networks, yielding a more brain like iterative computation characteristic of asymmetric biological neural circuits. Our approach thus avoids the biologically implausible requirement of *weight transport* (Lillicrap et al., 2014) which arises as a consequence of imposing weight symmetry as a hard constraint. With VW, this hard constraint is removed, although the training procedure itself may converge towards more symmetry. Such approach towards symmetry is consistent with both empirical observations (Vincent et al., 2010) and theoretical analysis (Arora et al., 2015) of auto-encoders, for which symmetric weights are associated with minimizing reconstruction error.

## 6.4 A connection to the neurobiology of dreams

The learning rule underlying VW, when applied to an asymmetric stochastic neural network, yields a speculative, but intriguing connection to the neurobiology of dreams. As discussed in Bengio et al. (2015), spike-timing dependent plasticity (STDP), a plasticity rule found in the brain (Markram and Sakmann, 1995), corresponds to increasing the probability of configurations towards which the network intrinsically likes to go (i.e., remembering observed configurations), while reverse-STDP corresponds to forgetting or unlearning the states towards which the network goes (which potentially may occur during sleep).

In the VW update applied to a neural network, the resultant learning rule does indeed strengthen synapses for which a presynaptic neuron is active before a postsynaptic neuron in the generative cooling process (STDP), and it weakens synapses in which a postsynaptic neuron is active before a presynaptic neuron in the heated destructive process (reverse STDP). If, as suggested, the neurobiological function of sleep involves re-organizing memories and in particular unlearning spurious modes through reverse-STDP, then the heating destructive process may map to sleep states, in which the brain is hunting down and destroying spurious modes. In contrast, the cooling generative dynamics of VW may map to awake states in which STDP reinforces neural trajectories moving towards observed sensory data. Under this mapping, the relative incoherence of dreams compared to reality is qualitatively consistent with the heated destructive dynamics of VW, compared to the cooled transition operator in place during awake states.

## 6.5 Future work

Many questions remain open in terms of analyzing and extending VW. Of particular interest is the incorporation of latent layers. The state at each step would now include both visible $x$ and latent $h$ components. Essentially the same procedure can be run, except for the chain initialization, with $s_0 = (x, h_0)$ where $h_0$ a sample from the posterior distribution of $h$ given $x$.

Another interesting direction is to replace the log-likelihood objective at each step by a GAN-like objective, thereby avoiding the need to inject noise independently on each of the pixels, during each transition step, and allowing latent variable sampling to inject the required high-level decisions associated with the transition. Based on the earlier results from (Bengio et al., 2013a), sampling in the latent space rather than in the pixel space should allow for better generative models and even better mixing between modes (Bengio et al., 2013a).

Overall, our work takes a step to filling a relatively open niche in the machine learning literature on *directly* training non-energy-based iterative stochastic operators, and we hope that the many possible extensions of this approach could lead to a rich new class of more powerful brain-like machine learning models.

## Acknowledgments

The authors would like to thank Benjamin Scellier, Ben Poole, Tim Cooijmans, Philemon Brakel, Gaétan Marceau Caron, and Alex Lamb for their helpful feedback and discussions, as well as NSERC, CIFAR, Google, Samsung, Nuance, IBM and Canada Research Chairs for funding, and Compute Canada for computing resources. S.G. would like to thank the Simons, McKnight, James S. McDonnell, and Burroughs Wellcome Foundations and the Office of Naval Research for support. Y.B would also like to thank Geoff Hinton for an analogy which is used in this work, while discussing contrastive divergence (personnal communication). The authors would also like to express debt of gratitude towards those who contributed to theano over the years (as it is no longer maintained), making it such a great tool.

## Footnotes

*A transition operator maps the previous-state distribution to a next-state distribution, and is implemented by a stochastic transformation which from the previous state of a Markov chain generates the next state

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
