[Supplementary Material · Variational_walkback_nips2017_supplementary.pdf]

# Supplementary Material

## 1 VW transition operators and their convergence

**Proposition 1.** *If $p$ has enough capacity, training data and training time, with slow enough annealing and a small departure from reversibility so $p$ can match $q$, then at convergence of VW training, the transition operator $p_T$ at $T = 1$ has the data generating distribution as its stationary distribution.*

*Proof.* With these conditions $p(s_0^{K+n})$ match $q(s_0^{K+n})$, where $q(s_0)$ is the data distribution. It means that $p(s_0)$ (the marginal at the last step of sampling) is the data distribution when running the annealed (cooling) trajectory for $K + n$ steps, for $n$ any integer between 0 and $N_1$, where the last $n + 1$ steps are at temperature 1. Since the last $n$ steps are at temperature 1, they apply the same transition operator. Consider any 2 consecutive sampling steps among these last $n$ steps. Both of these samples are coming from the same distribution (the data distribution). It means that the temperature 1 transition operator leaves the data distribution unchanged. This implies that the data distribution is an eigenvector of the linear operator associated with the temperature 1 transition operator, or that the data generating distribution is a stationary distribution of the temperature 1 transition operator. $\square$

## 2 Additional Results

Image inpainting samples from CelebA dataset are shown in Fig 1, where each top sub-figure shows the masked image of a face (starting point of the chain), and the bottom sub-figure shows the inpainted image. The images are drawn from the test set.

The VW samples for CelebA, CIFAR10 and SVHN are shown in Fig 3, 4, 5.

Figure 1: VW inpainting in CelebA images. Images on the left are the ground truth images corrupted for their bottom half (which is the starting point of the chain). The goal is to fill in the bottom half of each face image given an observed top half of an image (drawn from test set). Images on the right show the inpainted lower halves for all these images.

## 3 VW on Toy Datasets

Fig. 6 and 7 shows the application of a transition operator applied on 2D datasets.

Figure 2: VW samples on MNIST using Gaussian noise in the transition operator. The model is trained with 30 steps of walking away, and samples are generated using 30 annealing steps.

## 4 VW chains

Fig. 8, 9, 10, 11, 12, 13, 14 shows the model chains on repeated application of transition operator at temperature = 1. This is to empirically prove the conjecture mentioned in the paper (Preposition 1) that is, if the finite time generative process converges to the data distribution at multiple different VW walkback time-steps, then it remains on the data distribution for all future time at T= 1

## 5 Architecture Details

In this section, we provide more details on the architecture that was used for each of the dataset. The details of the hyper parameter and architecture used for each dataset can also be found in Tables 1, 2, 3 and 4. Complete specifications are available as experiment scripts at `http://github.com/anirudh9119/walkback_nips17`.

### 5.1 MNIST

For lower bound(and IS estimates) comparisons, the network trained on MNIST is a MLP composed of two fully connected layers with 1200 units using batch-normalization (Ioffe  Szegedy, 2015) This network has two different final layers with a number of units corresponding to the image size (i.e

Figure 3: VW samples on CelebA dataset using Gaussian noise in the transition operator. Model is trained using 30 steps to walk away and samples are generated using 30 annealing steps.

number of pixels) each corresponding to mean and variance for each pixel. We use softplus output for the variance. We don't share the batch-normalization parameters across different time steps.

For the real-values MNIST dataset samples, we used an encoder-decoder architecture with convolutional layers. The encoder consists of 2 convolutional layers with kernel length of 5 and stride of 2 followed by a decoder with strided convolutions. In addition, we used 5 fully connected feedforward layers to connect the encoder and decoder. We applied batch normalization (Ioffe and Szegedy, 2015) to the convolutional Layers, and we applied layer normalization (Ba et al., 2016) to the feedforward layers. The network has 2 separate output layers, one corresponding the mean of the Gaussian sample, and one corresponding to the variance of the added Gaussian noise. We use Adam (Kingma and Ba, 2014) with a learning rate of 0.0001 to optimize the network. Details of the hyper parameter and architecture is also available in Table 1.

## 5.2 CIFAR10, CelebA and SVNH

We use a similar encoder-decoder architecture as we have stated above. We use 3 convolutional layers for the encoder as well as for the decoder. We also apply batch normalization (Ioffe and Szegedy, 2015)to the convolutional layers, as well as layer normalization (Ba et al., 2016) to the feedforward layers. Details of the hyper parameter and architecture is also available in Table 3, 4 and 2.

| Operation | Kernel | Strides | Feature Maps | Normalization | Non Linearity | Hidden Units |
|---|---|---|---|---|---|---|
| Convolution | 5 x 5 | 2 | 16 | Batchnorm | Relu | - |
| Convolution | 5 x 5 | 2 | 32 | Batchnorm | Relu | - |
| Fully Connected | - | - | - | LayerNorm | Leaky Relu | 1568 * 1024 |
| Fully Connected | - | - | - | LayerNorm | Leaky Relu | 1024 * 1024 |
| Fully Connected | - | - | - | LayerNorm | Leaky Relu | 1024 * 1024 |
| Fully Connected | - | - | - | LayerNorm | Leaky Relu | 1024 * 1024 |
| Fully Connected | - | - | - | LayerNorm | Leaky Relu | 1024 * 1568 |
| Strided Convolution | 5 x 5 | 2 | 16 | Batchnorm | Relu | - |
| Strided Convolution | 5 x 5 | 2 | 1 | No | None | - |

Table 1: Hyperparameters for MNIST experiments, for each layer of the encoder-decoder (each row of the table). We use adam as an optimizer, learning rate of 0.0001. We model both mean and variance of each pixel. We use reconstruction error as per-step loss function. We see improvements using layernorm in the bottleneck, as compared to batchnorm. Using Dropout also helps, but all the results reported in the paper are without dropout.

| Operation | Kernel | Strides | Feature Maps | Normalization | Non Linearity | Hidden Units |
|---|---|---|---|---|---|---|
| Convolution | 5 x 5 | 2 | 64 | Batchnorm | Relu | - |
| Convolution | 5 x 5 | 2 | 128 | Batchnorm | Relu | - |
| Convolution | 5 x 5 | 2 | 256 | Batchnorm | Relu | - |
| Fully Connected | - | - | - | Batchnorm | Relu | 16384 * 1024 |
| Fully Connected | - | - | - | Batchnorm | Relu | 1024 * 1024 |
| Fully Connected | - | - | - | Batchnorm | Relu | 1024 * 1024 |
| Fully Connected | - | - | - | Batchnorm | Relu | 1024 * 1024 |
| Fully Connected | - | - | - | Batchnorm | Relu | 1024 * 16384 |
| Strided Convolution | 5 x 5 | 2 | 128 | Batchnorm | Relu | - |
| Strided Convolution | 5 x 5 | 2 | 64 | Batchnorm | Relu | - |
| Strided Convolution | 5 x 5 | 2 | 3 | No | None | - |

Table 2: Hyperparameters for CelebA experiments, for each layer of the encoder-decoder (each row of the table). We use adam as an optimizer, learning rate of 0.0001. We model both mean and variance of each pixel. We use reconstruction error as per-step loss function.

| Operation | Kernel | Strides | Feature Maps | Normalization | Non Linearity | Hidden Units |
|---|---|---|---|---|---|---|
| Convolution | 5 x 5 | 2 | 64 | Batchnorm | Relu | - |
| Convolution | 5 x 5 | 2 | 128 | Batchnorm | Relu | - |
| Convolution | 5 x 5 | 2 | 256 | Batchnorm | Relu | - |
| Fully Connected | - | - | - | Batchnorm | Relu | 4096 * 2048 |
| Fully Connected | - | - | - | Batchnorm | Relu | 2048 * 2048 |
| Fully Connected | - | - | - | Batchnorm | Relu | 2048 * 2048 |
| Fully Connected | - | - | - | Batchnorm | Relu | 2048 * 2048 |
| Fully Connected | - | - | - | Batchnorm | Relu | 2048 * 4096 |
| Strided Convolution | 5 x 5 | 2 | 128 | Batchnorm | Relu | - |
| Strided Convolution | 5 x 5 | 2 | 64 | Batchnorm | Relu | - |
| Strided Convolution | 5 x 5 | 2 | 3 | No | None | - |

Table 3: Hyperparameters for Cifar experiments, for each layer of the encoder-decoder (each row of the table). We use adam as an optimizer, learning rate of 0.0001. We model both mean and variance of each pixel. We use reconstruction error as per-step loss function.

Figure 4: VW samples on Cifar10 using Gaussian noise in the transition operator. Model is trained using 30 steps to walk away and samples are generated using 30 annealing steps.

## 6   Walkback Procedure Details

The variational walkback algorithm has three unique hyperparameters. We specify the number of Walkback steps used during training, the number of extra Walkback steps used during sampling and also the temperature increase per step.

The most conservative setting would be to allow the model to slowly increase the temperature during training. However, this would require a large number of steps for the model to walk to the noise, and this would not only significantly slow down the training process, but this also means that we would require a large number of steps used for sampling.

There may exist a dynamic approach for setting the number of Walkback steps and the temperature schedule. In our work, we set this hyperparameters heuristically. We found that a heating temperature schedule of $T_t = T_0\sqrt{2^t}$ at step $t$ produced good results, where $T_0 = 1.0$ is the initial temperature. During sampling, we found good results using the exactly reversed schedule: $T_t = \frac{\sqrt{2^N}}{\sqrt{2^t}}$, where $t$ is the step index and $N$ is the total number of cooling steps.

For MNIST, CIFAR, SVHN and CelelbA, we use $K = 30$ training steps and $N = 30$ sampling steps. We also found that we could achieve better quality results if allow the model to run for a few extra steps with a temperature of 1 during sampling. Finally, our model is able to achieve similar results compared to the NET model(Sohl-Dickstein et al., 2015). Considering our model uses only 30 steps for MNIST and NET (Sohl-Dickstein et al., 2015) uses 1000 steps for MNIST.

Figure 5: VW samples on SVHN dataset using Gaussian noise in the transition operator. Model is trained using 30 steps to walk away and samples are generated using 30 annealing steps.

## 7    Higher Lower Bound: not always better samples

We have observed empirically that the variational lower bound does not necessarily correspond to sample quality. Among trained models, higher value of the lower bound is not a clear indication of visually better looking samples. Our MNIST samples shown in Fig 15 is an example of this phenomenon. A model with better lower bound could give better reconstructions while not producing better generated samples. This resonates with the finding of (Theis et al., 2016)

## 8    Reversibility of transition operator

We measured the degree of reversibility of $p_T$ by estimating the KL divergence $D_{KL}(p_T(s'|s)\pi_T(s) \, \| \, p_T(s|s')\pi_T(s'))$, which is 0 if and only if $p_T$ obeys detailed balance and is therefore time-reversal invariant by computing the Monte-Carlo estimator $\frac{1}{K}\sum_{t=1}^{K} \ln \frac{p_T(s_{t+1}|s_t)}{p_T(s_t|s_{t+1})}$, where $s_1^K$ is a long sequence sampled by repeatedly applying transition operator $p_T$ from a draw $s_1 \sim \pi_T$, i.e., taking samples after a burn-in period (50 samples).

To get a sense of the magnitude of this reversibility measure, and because it corresponds to an estimated KL divergence, we estimate the corresponding entropy (of the forward trajectory) and use it as a normalizing denominator telling us how much we depart from reversibility in nats relative to the number of nats of entropy. To justify this, consider that the minimal code length required to code

Figure 6: The proposed modeling framework trained on 2-d swiss roll data. This algorithm was trained on 2D swiss roll for 30 annealing steps using annealing schedule increasing temperator by 1.1 each time. We have shown every 5th sample (ordering is row wise, and within each row it is column-wise.

Figure 7: The proposed modeling framework trained on circle data. This algorithm was trained on circle for 30 annealing time steps using annealing schedule increasing temperature by factor 1.1 each time. We have shown every 5th sample (ordering is row wise, and within each row it is column-wise.

85  samples from a distribution $p$ is the entropy $H(p)$. But suppose we evaluate those samples from $p$
86  using $q$ instead to code them. Then the code length is $H(p) + D(p||q)$. So the fractional increase
87  in code length due to having the wrong distribution is $D(p||q)/H(p)$, which is what we report here,
88  with $p$ being the forward transition probability and $q$ the backward transition probability.

89  To compute this quantity, we took our best model (in terms of best lower bound) on MNIST, and ran
90  it for 1000 time steps i.e ($T = 1000$), at a constant temperature.

91  We run the learned generative chain $p$ for $T$ time steps (after a burn in period
92  whose samples we ignore) getting $s_0 \rightarrow s_1 \rightarrow s_2 \rightarrow \cdots s_T$ and computing
93  $\log p(s_0 \rightarrow s_1 \rightarrow s_2 \rightarrow \cdots s_T)/p(s_T \rightarrow \cdots \rightarrow s2 \rightarrow s1)$ both under the same generative chain, di-
94  vided by $T$ to get the per-step average.

95  On the same set of runs, we compute $1/T * \log p(s_0 \rightarrow s_1 \rightarrow s_2 \rightarrow \cdots s_T)$ under the same generative
96  chain. This is an estimate of the entropy per unit time of the chain. This is repeated multiple times to
97  average over many runs and reduce the variance of the estimator.

Figure 8: VW sample chain (vertically, going down) starting from pure noise. Model trained using $K = 30$ steps to walk away and samples are generated using 30 steps of annealing. The figure shows every 3rd sample of the chain in each column.

The obtained ratio (nats/nats) is 3.6%, which seems fairly low but also suggests that the trained model is not perfectly reversible.

# 9 Some Minor Points

- In all the image experiments, we observed that by having different batchnorm papemeters for different steps, actually improves the result considerably. Having different batchnorm parameters was also necessery for making it work on mixture on gaussian. The authors were not able to make it work on MoG without different parameters. One possible way, could be to let optimizer know that we are on different step by giving the temperature information to the optimizer too.

- We observed better results while updating the parameters in online-mode, as compared to batch mode. (i.e instead of accumulating gradients across different steps, we update the parameters in an online fashion)

Figure 9: VW sample chain. Each coloumn above corresponds to one sampling chain. We have shown every 10th sample. We applied the transition operator for 5000 time-steps at temperature = 1, to demonstrate that even over very long chain, the transition operator continues to generate good samples.

## 10 Inception Scores on CIFAR

We computed the inception scores using 50,000 samples generated by our model. We compared the inception scores with (Salimans et al., 2016) as the baseline model.

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

Figure 10: VW sample chain. Each column above corresponds to one sampling chain. We have shown every 10th sample. We applied the transition operator for 5000 time-steps at temperature =

122  Sohl-Dickstein, J., Weiss, E. A., Maheswaranathan, N., and Ganguli, S. (2015). Deep unsupervised
123      learning using nonequilibrium thermodynamics. *CoRR*, abs/1503.03585.

124  Theis, L., van den Oord, A., and Bethge, M. (2016). A note on the evaluation of generative models.
125      In *International Conference on Learning Representations*.

Figure 11: VW sample chain. Each column above corresponds to one sampling chain. We have shown every 10th sample. We applied the transition operator for 5000 time-steps temperature = 1.

| Operation | Kernel | Strides | Feature Maps | Normalization | Non Linearity | Hidden Units |
|---|---|---|---|---|---|---|
| Convolution | 5 x 5 | 2 | 64 | Batchnorm | Relu | - |
| Convolution | 5 x 5 | 2 | 128 | Batchnorm | Relu | - |
| Convolution | 5 x 5 | 2 | 256 | Batchnorm | Relu | - |
| Fully Connected | - | - | - | Batchnorm | Relu | 4096 * 1024 |
| Fully Connected | - | - | - | Batchnorm | Relu | 1024 * 1024 |
| Fully Connected | - | - | - | Batchnorm | Relu | 1024 * 1024 |
| Fully Connected | - | - | - | Batchnorm | Relu | 1024 * 1024 |
| Fully Connected | - | - | - | Batchnorm | Relu | 1024 * 4096 |
| Strided Convolution | 5 x 5 | 2 | 128 | Batchnorm | Relu | - |
| Strided Convolution | 5 x 5 | 2 | 64 | Batchnorm | Relu | - |
| Strided Convolution | 5 x 5 | 2 | 3 | No | None | - |

Table 4: Hyperparameters for SVHN experiments, for each layer of the encoder-decoder (each row of the table). We use adam as an optimizer, learning rate of 0.0001. We model both mean and variance of each pixel. We use reconstruction error as per-step loss function.

Figure 12: VW sample chain. Each column above corresponds to one sampling chain. We have shown every 10th sample. We applied the transition operator for 5000 time-steps at temperature = 1, to demonstrate that even over very long chain, the transition operator continues to generate good samples.

| Model | Inception Score |
|---|---|
| Real Data | 11.24 |
| Salimans (semi-supervised) | 8.09 |
| Salimans (unsupervised) | 4.36 |
| Salimans (supervised training without minibatch features) | 3.87 |
| VW(20 steps) | 3.72 |
| VW(30 steps) | 4.39 $\pm$0.2 |

Table 5: Inception scores on CIFAR

Figure 13: VW sample chain. Each column above corresponds to one sampling chain. We have shown every 10th sample. We applied the transition operator for 5000 time-steps at temperature = 1, to demonstrate that even over very long chain, the transition operator continues to generate good samples.

Figure 14: VW sample chain. Each column above corresponds to one sampling chain. We have shown every 10th sample. We applied the transition operator for 5000 time-steps at temperature = 1, to demonstrate that even over very long chain, the transition operator continues to generate good samples.

Figure 15: Samples from two VW models (left and right) which have a higher lower bound than the one whose samples are shown in Figure 5 (and comparable but slightly better importance sampling estimators of the log-likelihood): yet, the generated samples are clearly not as good, suggesting that either the bound is sometimes not tight enough or that the log-likelihood is not always a clear indicator of sample quality.