[Reviews · NeurIPS 2017]

Reviewer 1



This paper proposes an extension of the information destruction/reconstruction processes introduced by Sohl-Dickstein et al. (i.e. NET). Specifically, the authors propose learning an explicit model for the information destroying process, as opposed to defining it a priori as repeated noising and rescaling. The authors also propose tying the parameters of the forwards/backwards processes, with the motivation of more efficiently seeking to eliminate spurious modes (kind of like contrastive divergence in undirected, energy-based models). The general ideas explored in this paper are interesting -- i.e. training models which generate data through stochastic iterative refinement, and which provide mechanisms for "wandering around" the data manifold. However, the proposed model is a fairly straightforward extension of NET and does not produce compelling quantitative or qualitative results. The general perspective taken in this paper, i.e. reinterpreting the NET approach in the context of variational inference, was presented earlier in Section 2.3 of "Data Generation as Sequential Decision Making" by Bachman et al. (NIPS 2015). If more ambitious extensions of the ideas latent in NET were explored in this paper, I would really like it. But, I don't see that yet. And, e.g., the CIFAR10 results are roughly what one gets when simply modelling the data using a full-rank Gaussian distribution. This would make an interesting workshop paper, but I don't think there's enough strong content yet to fill a NIPS paper.

Reviewer 2



This is an interesting paper that presents a new generative modeling idea using stochastic recurrent nets. It is inspired by the physical annealing process to obtain a transition operator that can "walk-back" to the original data distribution. The tightness of the variational bounds is important. While it presents some quantitative results on a few data sets and compared with a few other generative models, a little more discussion of the results would be useful in the main paper. The overall paper is also a little difficult to understand at times, so, the writing can be improved for better readability.

Reviewer 3



The paper proposes an exciting practical and theoretical framework for unsupervised estimation of distribution. The framework is reminiscent of Generalized Denoising Auto-Encoders and likewise proposes to train the transition operator in a Markov Chain (e.g. a deep network). The performance of the method is evaluated on several datasets and is impressive although maybe not ground breaking. On the other hand, the theoretical framework connects many existing models in very interesting ways and offers interesting insights as to their relationship. The model also gives a sound solution to sampling with a transition operator which does not respect the detailed-balance condition. This is a very important insight that will probably be very helpful to theoreticians and practitionners alike.